# Reward Uncertainty for Exploration in Preference-based Reinforcement Learning

**Xinran Liang**[1] **, Katherine Shu**[1] **, Kimin Lee**[1]* **, Pieter Abbeel**[1]*
[1]University of California, Berkeley

## Abstract

Conveying complex objectives to reinforcement learning (RL) agents often requires meticulous reward engineering. Preference-based RL methods are able to learn a more flexible reward model based on human preferences by actively incorporating human feedback, i.e. teacher's preferences between two clips of behaviors. However, poor feedback-efficiency still remains a problem in current preference-based RL algorithms, as tailored human feedback is very expensive. To handle this issue, previous methods have mainly focused on improving query selection and policy initialization. At the same time, recent exploration methods have proven to be a recipe for improving sample-efficiency in RL. We present an exploration method specifically for preference-based RL algorithms. Our main idea is to design an intrinsic reward by measuring the novelty based on learned reward. Specifically, we utilize disagreement across ensemble of learned reward models. Our intuition is that disagreement in learned reward model reflects uncertainty in tailored human feedback and could be useful for exploration. Our experiments show that exploration bonus from uncertainty in learned reward improves both feedback- and sample-efficiency of preference-based RL algorithms on complex robot manipulation tasks from MetaWorld benchmarks, compared with other existing exploration methods that measure the novelty of state visitation.

## 1 Introduction

In reinforcement learning (RL), reward function specifies correct objectives to RL agents. However, it is difficult and time-consuming to carefully design suitable reward functions for a variety of complex behaviors (e.g., cooking or book summarization (Wu et al., 2021)). Furthermore, if there are complicated social norms we want RL agents to understand and follow, conveying a reliable reward function to include such information may remain to be an open problem (Amodei et al., 2016; Hadfield-Menell et al., 2017). Overall, engineering reward functions purely by human efforts for all tasks remains to be a significant challenge.

An alternative to resolve the challenge of reward engineering is preference-based RL (Christiano et al., 2017; Ibarz et al., 2018; Lee et al., 2021b). Compared to traditional RL setup, preference-based RL algorithms are able to teach RL agents without the necessity of designing reward functions. Instead, the agent uses feedback, usually in the form of (human) teacher preferences between two behaviors, to learn desired behaviors indicated by teacher. Therefore, instead of using carefully-designed rewards from the environment, the agent is able to learn a more flexible reward function suitably aligned to teacher feedback.

However, preference-based RL usually requires a large amount of teacher feedback, which may be timely or sometimes infeasible to collect. To improve feedback-efficiency, prior works have investigated several sampling strategies (Biyik & Sadigh, 2018; Sadigh et al., 2017; Biyik et al., 2020; Lee et al., 2021c). These methods aim to select more informative queries to improve the quality of the learned reward function while asking for fewer feedback from teacher. Another line of works focus on policy initialization. Ibarz et al. (2018) initialized the agent's policy with imitation learning from the expert demonstrations, and Lee et al. (2021b) utilized unsupervised pre-training of RL agents before collecting for teacher preferences in the hope of learning diverse behaviors in a self-supervised way to reduce total amount of human feedback.

---

* Equal advising.

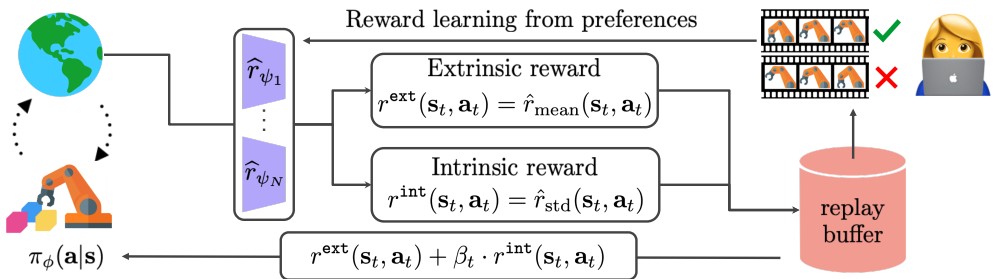

Figure 1: Illustration of RUNE. The agent interacts with the environment and learns an ensemble of reward functions based on teacher preferences. For each state-action pair, the total reward is a combination of the extrinsic reward, the mean of the ensemble's predicted values, and the intrinsic reward, the standard deviation between the ensemble's predicted values.

Exploration, in the context of standard RL, has addressed the problems of sample-efficiency (Stadie et al., 2015; Bellemare et al., 2016; Pathak et al., 2017; 2019; Liu & Abbeel, 2021; Seo et al., 2021b). When extrinsic rewards from the environment is limited, exploration has been demonstrated to allow RL agents to learn diverse behaviors. However, limited previous works have studied the effects of exploration in preference-based RL.

Inspired by recent success of exploration methods, we present RUNE: **R**eward **UN**certainty for **E**xploration, a simple and efficient exploration method specifically for preference-based RL algorithms. Our main idea is to incorporate uncertainty from learned reward function as an exploration bonus. Specifically, we capture the novelty of human feedback by measuring the reward uncertainty (e.g., variance in predictions of ensemble of reward functions). Since reward functions is optimized and learned to align to human feedback, exploration based on reward uncertainty may also reflect high uncertainty in information from teacher feedback. We hope that the proposed intrinsic reward contains information from teacher feedback and can guide exploration that better align to human preferences. Our experiment results show that RUNE can improve both sample- and feedback-efficiency of preference-based RL algorithms (Lee et al., 2021b).

We highlight the main contributions of our paper below:

- We propose a new exploration method based on uncertainty in learned reward functions for preference-based RL algorithms.

- For the first time, we show that *exploration* can improve the sample- and feedback-efficiency of preference-based RL algorithms.

## 2   RELATED WORK

**Human-in-the-loop reinforcement learning**. We mainly focus on one promising direction that utilizes the human preferences (Akrour et al., 2011; Christiano et al., 2017; Ibarz et al., 2018; Lee et al., 2021b; Leike et al., 2018; Pilarski et al., 2011; Wilson et al., 2012) to train RL agents. Christiano et al. (2017) scaled preference-based learning to utilize modern deep learning techniques, and Ibarz et al. (2018) improved the efficiency of this method by introducing additional forms of feedback such as demonstrations. Recently, Lee et al. (2021b) proposed a feedback-efficient RL algorithm by utilizing off-policy learning and pre-training.

To improve sample- and feedback-efficiency of human-in-the-loop RL, previous works (Christiano et al., 2017; Ibarz et al., 2018; Lee et al., 2021b; Leike et al., 2018) mainly focus on methods such as selecting more informative queries Christiano et al. (2017) and pre-training of RL agents Ibarz et al. (2018); Lee et al. (2021b). We further investigate effects of different exploration methods in preference-based RL algorithm. We follow a common approach of exploration methods in RL: generating intrinsic rewards as exploration bonus Pathak et al. (2019). Instead of only using learned reward function from human feedback as RL training objective, we alter the reward function to include a combination of the extrinsic reward (the learned rewards) and an intrinsic reward (explo-

ration bonus). In particular, we present an exploration method with intrinsic reward that measures the disagreement from learned reward models.

**Exploration in reinforcement learning.** The trade off between exploitation and exploration is a critical topic in RL. If agents don't explore enough, then they may learn sub optimal actions. Exploration algorithms aim to encourage the RL agent to visit a wide range of states in the environment. Thrun (1992) showed that exploration methods that utilize the agent's history has been shown to perform much better than random exploration. Hence, a common setup is to include an intrinsic reward as an exploration bonus. The intrinsic reward can be defined by Count-Based methods which keep count of previously visited states and rewards the agents for visiting new states Bellemare et al. (2016); Tang et al. (2017); Ostrovski et al. (2017).

Another option is to use a curiosity bonus for the intrinsic reward Houthooft et al. (2016); Pathak et al. (2017); Sekar et al. (2020). Curiosity represents how expected and unfamiliar the state is. One way to quantify curiosity is to predict the next state from current state and action Pathak et al. (2017), then use prediction error as an estimate of curiosity. If the error is high, that means the next state is unfamiliar and should be explored more. Similarly, instead of predicting the next state, prediction errors from training a neural network to approximate a random function Burda et al. (2018) can serve as a valid estimate of curiosity. If there are multiple models, then curiosity can also be described as the disagreement between the models Pathak et al. (2019). A high disagreement means that the models are unsure about the prediction and need to explore in that direction more.

A different approach maximizes the entropy of visited states by incorporating state entropy into the intrinsic reward. State entropy can be estimated by approximating the state density distribution Hazan et al. (2019); Lee et al. (2019), approximating the k-nearest neighbor entropy of a randomly initialized encoder Seo et al. (2021a), or using off-policy RL algorithms to maximize the $k$-nearest neighbor state entropy estimate in contrastive representation space for unsupervised pre-training Srinivas et al. (2020); Liu & Abbeel (2021). These methods encourage agents to explore diverse states.

Our approach adds an intrinsic reward that drives exploration to preference-based RL algorithms. We take advantage of an ensemble of reward models in preference-based RL algorithms, which is not available in other traditional RL settings. To estimate novelty of states and actions, we utilize the disagreement between reward models for our intrinsic reward, in hope of encouraging exploration aligned to directions of human preferences.

**Trajectory generation in preference-based reinforcement learning.** Previous works in preference-based reinforcement learning have investigated several methods to better explore diverse trajectories but close to current optimal policy Wirth et al. (2017).

One line of works computes agent's stochastic policies that are slightly deviated from optimal policies. Christiano et al. (2017) uses Trust Region Policy Optimization (TRPO) Schulman et al. (2015) and synchronized A3C Mnih et al. (2016). These RL algorithms define stochastic policies to ensure exploration of action space and deviations from optimal policies. However, these exploration methods based on stochastic RL algorithms does not include information from human preferences to drive exploration.

Another line of works designs one or multiple criterion to select from multiple possible stochastic policy candidates. Wilson et al. (2012) proposes to sample several policies from posterior distribution of policy space after updating human preferences. However, such methods come a the cost of requiring many samples collected beforehand. While these methods similarly aims to reduce uncertainty in human preferences, RUNE uses a different metric to estimate such uncertainty through reward functions ensemble. This is different from previous works and is simple, scalable, and easy to implement.

A different approach allows human to guide exploration by directly providing additional trajectories. Zucker et al. (2010) proposes a user-guided exploration method that shows samples of trajectories to human. Human can provide additional feedback to guide exploration. While this method receives exact information from human, it requires additional human labels, which are usually expensive and time-consuming to collect. RUNE however tries to extract information from human feedback revealed in learned reward functions, which doesn't require additional human input.

## 3 PRELIMINARIES

**Preference-based reinforcement learning**. We consider an agent interacting with an environment in discrete time Sutton & Barto (2018). At each timestep $t$, the agent receives a state $\mathbf{s}_t$ from the environment and chooses an action $\mathbf{a}_t$ based on its policy $\pi$.

In traditional reinforcement learning, the environment also returns a reward $r(\mathbf{s}_t, \mathbf{a}_t)$ that evaluates the quality of agent's behavior at timestep $t$. The goal of agent is to maximize the discounted sum of rewards. However, in the preference-based RL framework, we don't have such a reward function returned from the environment. Instead, a (human) teacher provides preferences between the agent's behaviors and the agent learns its policy from feedbacks (Christiano et al., 2017; Ibarz et al., 2018; Lee et al., 2021b; Leike et al., 2018).

Formally, a segment $\sigma$ is a sequence of time-indexed observations and actions $\{(\mathbf{s}_1, \mathbf{a}_1), ..., (\mathbf{s}_H, \mathbf{a}_H)\}$. Given a pair of segments $(\sigma^0, \sigma^1)$ that describe two behaviors, a teacher indicates which segment is preferred, i.e., $y = (\sigma^0 \succ \sigma^1)$ or $(\sigma^1 \succ \sigma^0)$, that the two segments are equally preferred $y = (\sigma^1 = \sigma^0)$, or that two segments are incomparable, i.e., discarding the query. The goal of preference-based RL is to train an agent to perform behaviors desirable to a human teacher using as few feedback as possible.

In preference-based RL algorithms, a policy $\pi_\phi$ and reward function $\widehat{r}_\psi$ are updated as follows:

- *Step 1 (agent learning)*: The policy $\pi_\phi$ interacts with environment to collect experiences and we update it using existing RL algorithms to maximize the sum of the learned rewards $\widehat{r}_\psi$.

- *Step 2 (reward learning)*: The reward function $\widehat{r}_\psi$ is optimized based on the feedback received from a teacher.

- Repeat *Step 1* and *Step 2*.

To incorporate human preferences into *reward learning*, we optimize reward function $\widehat{r}_\psi$ as follows. Following the Bradley-Terry model (Bradley & Terry, 1952), we first model preference predictor of a pair of segments based on reward function $\widehat{r}_\psi$ as follows:

$$P_\psi[\sigma^1 \succ \sigma^0] = \frac{\exp \sum_t \widehat{r}_\psi(\mathbf{s}_t^1, \mathbf{a}_t^1)}{\sum_{i \in \{0,1\}} \exp \sum_t \widehat{r}_\psi(\mathbf{s}_t^i, \mathbf{a}_t^i)}, \tag{1}$$

where $\sigma^i \succ \sigma^j$ denotes the event that segment $i$ is preferable to segment $j$. Here, the intuition is that segments with more desirable behaviors should have higher predicted reward values from $\widehat{r}_\psi$. To align preference predictors of $\widehat{r}_\psi$ with labels received from human preferences, preference-based RL algorithms translate updating reward function to a binary classification problem. Specifically, the reward function $\widehat{r}_\psi$ parametrized by $\psi$ is updated to minimize the following cross-entropy loss:

$$\mathcal{L}^{\texttt{Reward}} = - \mathop{\mathbb{E}}_{(\sigma^0, \sigma^1, y) \sim \mathcal{D}} \left[ \mathbb{I}\{y = (\sigma^0 \succ \sigma^1)\} \log P_\psi[\sigma^0 \succ \sigma^1] + \mathbb{I}\{y = (\sigma^1 \succ \sigma^0)\} \log P_\psi[\sigma^1 \succ \sigma^0] \right]. \tag{2}$$

where we are given a set $D$ of segment pairs and corresponding human preferences.

Once reward function $\widehat{r}_\psi$ has been optimized from human preferences, preference-based RL algorithms train RL agents with any existing RL algorithms, treating predicted rewards from $\widehat{r}_\psi$ as reward function returned from the environment.

## 4 RUNE

In this section, we present RUNE: **R**eward **UN**certainty for **E**xploration (Figure 1), which encourages human-guided exploration for preference-based RL. The key idea of RUNE is to incentivize exploration by providing an intrinsic reward based on reward uncertainty. Our main hypothesis is that the reward uncertainty captures the novelty of human feedback, which can lead to useful behaviors for preference-based RL.

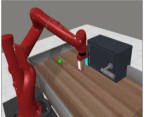 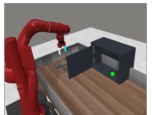 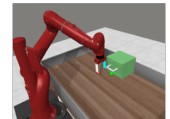 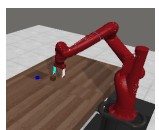 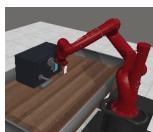 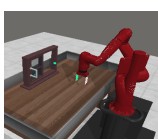

(a) Door Close    (b) Door Open    (c) Drawer Open    (d) Sweep Into    (e) Door Unlock    (f) Window Close

Figure 2: Examples of rendered images of robotic manipulation tasks from Meta-world. We consider learning several manipulation skills using preferences from a scripted teacher.

## 4.1 REWARD UNCERTAINTY FOR EXPLORATION

In preference-based RL, capturing the novelty of human feedback can be crucial for efficient reward learning. To this end, we propose to utilize an intrinsic reward based on ensemble of reward functions. Specifically, for each timestep, the intrinsic reward is defined as follows:

$$r^{\texttt{int}}(\mathbf{s}_t, \mathbf{a}_t) := \widehat{r}_{\text{std}}(\mathbf{s}_t, \mathbf{a}_t), \tag{3}$$

where $\widehat{r}_{\text{std}}$ is the empirical standard deviation of all reward functions $\{\widehat{r}_{\psi_i}\}_{i=1}^N$. We initialize the model parameters of all reward functions with random parameter values for inducing an initial diversity. Here, our intuition is high variance of reward functions indicates high uncertainty from human preferences, which means we collected less teacher preferences on those states and actions. Therefore, in order to generate more informative queries and improve confidence in learned reward functions, we encourage our agent to visit more uncertain state-action pairs with respect to the learned reward functions.

We remark that exploration based on ensembles has been studied in the literature (Osband et al., 2016; Chen et al., 2017; Pathak et al., 2019; Lee et al., 2021a). For example, Chen et al. (2017) proposed an exploration strategy that considers both best estimates (i.e., mean) and uncertainty (i.e., variance) of Q-functions and Pathak et al. (2019) utilized the disagreement between forward dynamics models. However, our method is different in that we propose an alternative intrinsic reward based on reward ensembles, which can capture the uncertainty from human preferences.

## 4.2 TRAINING OBJECTIVE BASED ON INTRINSIC REWARDS

Once we learn reward functions $\{\widehat{r}_{\psi_i}\}_{i=1}^N$ from human preferences, agent is usually trained with RL algorithm guided by extrinsic reward:

$$r^{\texttt{ext}}(\mathbf{s}_t, \mathbf{a}_t) = \widehat{r}_{\text{mean}}(\mathbf{s}_t, \mathbf{a}_t), \tag{4}$$

where $\widehat{r}_{\text{mean}}$ is the empirical mean of all reward functions $\{\widehat{r}_{\psi_i}\}_{i=1}^N$. To encourage exploration, we train a policy to maximize the sum of both extrinsic reward and intrinsic reward in equation 3:

$$r_t^{\texttt{total}} := r^{\texttt{ext}}(\mathbf{s}_t, \mathbf{a}_t) + \beta_t \cdot r^{\texttt{int}}(\mathbf{s}_t, \mathbf{a}_t), \tag{5}$$

where $\beta_t \geq 0$ is a hyperparameter that determines the trade off between exploration and exploitation at training time step $t$. Similar to Seo et al. (2021b), we use an exponential decay schedule for $\beta_t$ throughout training to encourage the agent to focus more on extrinsic reward from learned reward function predictions as training proceeds, i.e., $\beta_t = \beta_0(1 - \rho)^t$, where $\rho$ is a decay rate.

While the proposed intrinsic reward would converge to 0 as more feedback queries are collected during training, we believe our learned reward functions improve over training as we collect more feedback queries from teacher preferences. The full procedure of RUNE is summarized in Algorithm 1.

## 5 EXPERIMENTS

We designed our experiments to answer the following questions:

- Can exploration methods improve the sample- and feedback-efficiency of preference-based RL algorithms?

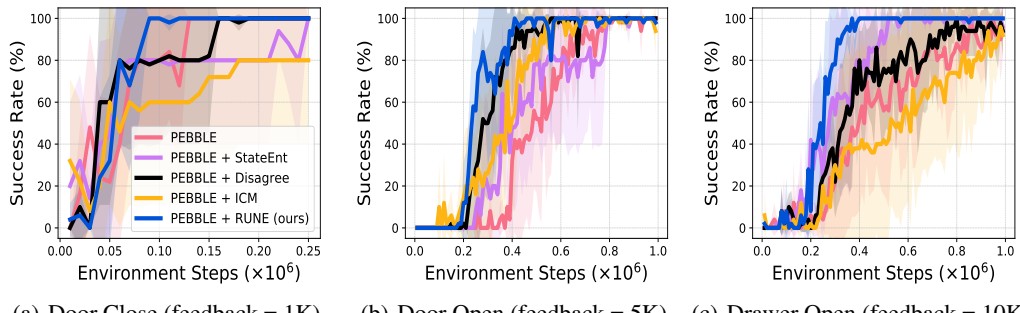

(a) Door Close (feedback = 1K)    (b) Door Open (feedback = 5K)    (c) Drawer Open (feedback = 10K)

Figure 3: Learning curves on robotic manipulation tasks as measured on the success rate. Exploration methods consistently improves the sample-efficiency of PEBBLE. In particular, RUNE provides larger gains than other existing exploration baselines. The solid line and shaded regions represent the mean and standard deviation, respectively, across five runs.

- How does RUNE compare to other exploration schemes in preference-based RL setting?
- How does RUNE influence reward learning in preference-based RL?

## 5.1 SETUP

In order to verify the efficacy of exploration in preference-based RL, we focus on having an agent solve a range of complex robotic manipulation skills from Meta-World (Yu et al., 2020) (see Figure 2). Similar to prior works (Christiano et al., 2017; Lee et al., 2021b;c), the agent learns to perform a task only by getting feedback from an oracle scripted teacher that provides clean preferences between trajectory segments according to the sum of ground truth reward values for each trajectory segment. Because the simulated human teacher's preferences are generated by a ground truth reward values, we measure the true average return of trained agents as evaluation metric.

We consider a combination of RUNE and PEBBLE (Lee et al., 2021b), an off-policy preference-based RL algorithm that utilizes unsupervised pre-training and soft actor-critic (SAC) method (Haarnoja et al., 2018) (see Appendix A for more details about algorithm procedure and Appendix B for more experimental details). To address the issue that there can be multiple reward functions that align to the same set of teacher preferences, we bound all predicted reward values to a normalized range of (-1, 1) by adding tanh activation function after the output layer in network architectures.

For all experiments including PEBBLE and RUNE, we train an ensemble of 3 reward functions according to Equation 2. Specifically, we use different random initialization for each network in the ensemble. We use the same set of training data, i.e. sampled set of feedback queries, but different random batches to train each network in the ensemble. Parameters of each network are independently optimized to minimize the cross entropy loss of their respective batch of training data according to Equation 2. In addition, we use original hyperparameters of preference-based RL algorithm, which are specified in Appendix B and report mean and standard deviations across 5 runs respectively.

## 5.2 IMPROVING SAMPLE-EFFICIENCY

To evaluate sample-efficiency of our method, we compare to the following exploration methods:

- State entropy maximization (StateEnt; Mutti et al. 2021; Liu & Abbeel 2021): Maximizing the entropy of state visitation distribution $H(\mathbf{s})$. We utilize a particle estimator (Singh et al., 2003), which approximates the entropy by measuring the distance between k-nearest neighbors ($k$-NN) for each state.
- Disagreement Pathak et al. (2019): Maximizing disagreement proportional to variance in predictions from ensembles $Var\{g_i(\mathbf{s}_{t+1}|\mathbf{s}_t, \mathbf{a}_t)\}_{i=1}^N$. We train an en-

semble of forward dynamics models to predict ground truth next state from current state and action $g_i(\mathbf{s}_{t+1}|\mathbf{s}_t, \mathbf{a}_t)$ by minimizing sum of prediction errors, i.e. $\sum_{i=1}^{N} ||g_i(\mathbf{s}_{t+1}|\mathbf{s}_t, \mathbf{a}_t) - s_{t+1}||_2$.

- ICM Pathak et al. (2017): Maximizing intrinsic reward proportional to prediction error $||g(\mathbf{s}_{t+1}|\mathbf{s}_t, \mathbf{a}_t) - \mathbf{s}_{t+1}||_2$. We train a single dynamics model to predict ground truth next state from current state and action $g(\mathbf{s}_{t+1}|\mathbf{s}_t, \mathbf{a}_t)$ via regression.

For all methods we consider, we carefully tune a range of hyperparameters and report the best results. In particular, we consider $\beta_0 = 0.05$ and $\rho \in \{0.001, 0.0001, 0.00001\}$ for all exploration methods, and $k \in \{5, 10\}$ for state entropy based exploration. We provide more details about training and implementation in Appendix B.

Figure 3 shows the learning curves of PEBBLE with various exploration methods. First, we remark that previous exploration methods (i.e., StateEnt, Disagree and ICM), which encourage agents to visit novel states, consistently improve sample-efficiency of PEBBLE on various tasks. This shows the potential and efficacy of exploration methods to further improve sample-efficiency of preference-based RL algorithms.

In particular, in Figure 3, compared to existing previous exploration methods, RUNE consistently exhibits superior sample efficiency in all tasks we consider. This suggests that exploration based on reward uncertainty is suitable for preference-based RL algorithms, as such it encourages visitations to states and actions that are uncertain with respect to human feedback and thus can capture the novelty in a distinctive perspective. We provide additional experiment results about better sample-efficiency in Figure 9(a) of Appendix C.

We also emphasize the simplicity and efficiency of RUNE compared to other existing schemes (such as ICM and Disagree) because our method does not introduce additional model architectures (e.g., ensemble of forward dynamics models) to compute exploration bonus as intrinsic rewards.

## 5.3 IMPROVING FEEDBACK-EFFICIENCY

In this section, we also verify whether our proposed exploration method can improve the feedback-efficiency of both off-policy and on-policy preference-based RL algorithms. We consider PEBBLE Lee et al. (2021b), an off-policy preference-based RL algorithm that utilizes unsupervised pre-training and soft actor-critic (SAC) method (Haarnoja et al., 2018), and PrefPPO Lee et al. (2021c), an on-policy preference-based RL algorithm that utilizes unsupervised pre-training and proximal policy gradient (PPO) method (Schulman et al., 2017), respectively.

As shown in Table 1, we compare performance of PEBBLE and PrefPPO with and without RUNE respectively using different budgets of feedback queries during training. With fewer total queries, we stop asking for human preferences earlier in the middle of training. We use asymptotic success rate converged at the end of training steps as evaluation metric. Table 1 suggests that RUNE achieves consistently better asymptotic performance using fewer number of human feedback; additionally RUNE shows more robust performance with respect to different budgets of available human feedbacks. This shows potential of exploration in scaling preference-based RL to real world scenarios where human feedback are usually expensive and time-consuming to obtain.

We report corresponding learning curves to show better sample efficiency of RUNE compared to PEBBLE or PrefPPO under wide variety of tasks environments in Figure 6, 7, 8, 5 of Appendix C.

## 5.4 ABLATION STUDY

**Ensemble size.** As intrinsic rewards of RUNE only depend on learned reward functions, we investigate the effects of different number of reward function ensembles. In particular, we consider $\{3, 5, 7\}$ number of reward function ensembles. In Figure 4(a), sample-efficiency of RUNE remains robust and stable with different number in the ensemble of reward functions. Additionally, RUNE can achieve better sample-efficiency in preference-based RL while requiring fewer number of reward functions ensembles. This shows potentials of RUNE specifically suitable for preference-based RL in improving compute efficiency, as it reduces the necessity of training additional reward functions architectures and still achieves comparable performance.

| Task | Feedback Queries | Method | Convergent Success Rate |
|---|---|---|---|
| Drawer Open | 10000 | PEBBLE | $0.98 \pm 0.08$ |
| | | PEBBLE + RUNE | $\mathbf{1 \pm 0}$ |
| | 5000 | PEBBLE | $0.94 \pm 0.08$ |
| | | PEBBLE + RUNE | $\mathbf{0.99 \pm 0.02}$ |
| Sweep Into | 10000 | PEBBLE | $0.8 \pm 0.4$ |
| | | PEBBLE + RUNE | $\mathbf{1 \pm 0}$ |
| | 5000 | PEBBLE | $0.8 \pm 0.08$ |
| | | PEBBLE + RUNE | $\mathbf{0.9 \pm 0.14}$ |
| Door Unlock | 5000 | PEBBLE | $0.66 \pm 0.42$ |
| | | PEBBLE + RUNE | $\mathbf{0.8 \pm 0.4}$ |
| | 2500 | PEBBLE | $0.64 \pm 0.45$ |
| | | PEBBLE + RUNE | $\mathbf{0.8 \pm 0.4}$ |
| Door Open | 4000 | PEBBLE | $1 \pm 0$ |
| | | PEBBLE + RUNE | $1 \pm 0$ |
| | 2000 | PEBBLE | $0.9 \pm 0.2$ |
| | | PEBBLE + RUNE | $\mathbf{1 \pm 0}$ |
| Door Close | 1000 | PEBBLE | $1 \pm 0$ |
| | | PEBBLE + RUNE | $1 \pm 0$ |
| | 500 | PEBBLE | $0.8 \pm 0.4$ |
| | | PEBBLE + RUNE | $\mathbf{1 \pm 0}$ |
| Window Close | 1000 | PEBBLE | $0.94 \pm 0.08$ |
| | | PEBBLE + RUNE | $\mathbf{1 \pm 0}$ |
| | 500 | PEBBLE | $0.86 \pm 0.28$ |
| | | PEBBLE + RUNE | $\mathbf{0.99 \pm 0.02}$ |
| Button Press | 20000 | PrefPPO | $0.46 \pm 0.20$ |
| | | PrefPPO + RUNE | $\mathbf{0.64 \pm 0.18}$ |
| | 10000 | PrefPPO | $0.35 \pm 0.31$ |
| | | PrefPPO + RUNE | $\mathbf{0.51 \pm 0.27}$ |

Table 1: Success rate of off- and on-policy preference-based RL algorithms (e.g. PEBBLE and PrefPPO) in addition to RUNE with different budgets of feedback queries. The results show the mean averaged and standard deviation computed over five runs and the best results are indicated in bold. All learning curves (including means and standard deviations) are in Appendix C.

**Number of queries per feedback session.** Human preferences are usually expensive and time-consuming to collect. As shown in Table 1, RUNE is able to achieve better asymptotic performance under different budgets of total human preferences. We further investigate the effects of different queries in each feedback session on performance of PEBBLE and RUNE. In particular, we consider $\{10, 50\}$ number of queries per feedback session equally spread out throughout training. In Figure 4(b), we indicate asymptotic performance of PEBBLE baseline by dotted horizontal lines. It shows that as $80\%$ of feedbacks are eliminated, asymptotic success rate of PEBBLE baseline largely drops, while in contrast RUNE remains superior performance in both sample-efficiency and asymptotic success. Thus RUNE is robust to different number of available feedback queries constraints and thus a suitable exploration method specifically beneficial for preference-based RL.

**Quality of learned reward functions.** We use Equivalent-Policy Invariant Comparison (EPIC) distance Gleave et al. (2020) between learned reward functions and ground truth reward values as evaluation metrics. Gleave et al. (2020) suggests that EPIC distance is robust to coverage distributions of states and actions and is thus a reliable metric to quantify distance between different reward functions under the same transition dynamics. We generate $(\mathbf{s}, \mathbf{a})$ from each task uniformly and independently at random in order to obtain a wide coverage of state distributions.

As in Figure 4(c), compared to PEBBLE, reward functions learned from RUNE are closer to ground truth reward evaluated based on EPIC distance and converges faster during training. We provide additional analysis on different task environments in Figure 9(b) and 9(c) of Appendix C.

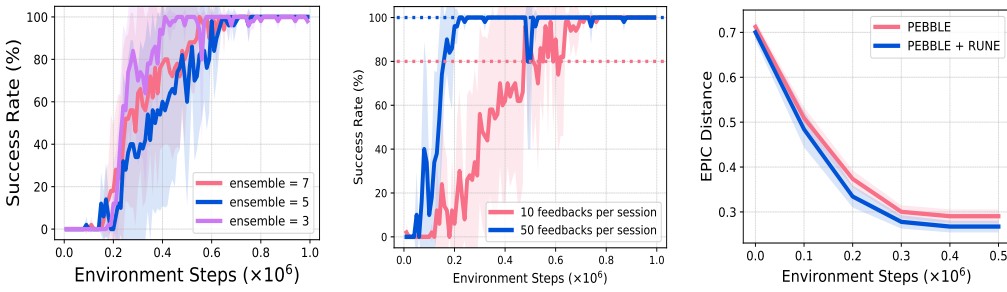

(a) Number of reward functions    (b) Number of queries per feedback (c) EPIC distance between learned
                                           session                         reward and true reward

Figure 4: Ablation study on (a) Door Open, (b) Window Close, and (c) Door Close. We measure (a) Effects of different number of learned reward model ensembles on RUNE; (b) Effects of different number of queries received per feedback session on PEBBLE and RUNE; and (c) EPIC distance between the ensemble of learned reward functions and true reward in the task environment. The solid line and shaded regions represent the mean and standard deviation, respectively, across five runs. The dotted line in Figure 4(b) indicates asymptotic success rate of PEBBLE baseline with same hyperparameters.

We further analyze quality of reward functions learned by RUNE compared to true reward in Figure 10 of Appendix C. Learned rewards optimized by RUNE is well-aligned to true reward, as it can capture diverse patterns of true reward values from the environment.

## 6   DISCUSSION

In this paper, we present RUNE, a simple and efficient exploration method for preference-based RL algorithms. To improve sample- and feedback-efficiency of preference-based RL, different from previous works, we investigate the benefits of exploration methods in preference-based RL. We show the significant potential of incorporating intrinsic rewards to drive exploration because it improves sample-efficiency of preference-based RL.

For our proposed exploration scheme RUNE, we show that it is useful for preference-based RL because it showcases consistently superior performance in both sample- and feedback-efficiency compared to other existing exploration methods. Here we emphasize that RUNE takes advantage of information in reward functions learned from human feedback, to measure the novelty of states and actions for exploration. This is different from existing estimates of uncertainty for exploration, as our method in particular encourages exploration aligned to teacher preferences.

In conclusion, we hope that our work could demonstrate the potential of exploration to improve sample- and feedback-efficiency of preference-based RL, and to encourage future works to develop novel exploration methods guided by human feedback.

## ACKNOWLEDGEMENTS

This research is supported in part by Open Philanthropy. We thank anonymous reviewers for critically reading the manuscript and suggesting substantial improvements.

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

# Appendix

## A    ALGORITHM DETAILS

**RUNE.** We provide the full procedure for RUNE with PEBBLE Lee et al. (2021b), an off-policy preference-based RL algorithm, in Algorithm 1.

---

**Algorithm 1** RUNE: Reward Uncertainty for Exploration

---

1: Initialize policy $\pi_\phi$ and reward model $\widehat{r}_\psi$
2: Initialize replay buffer $B$
3: // UNSUPERVISED PRE-TRAINING
4: Maximize the state entropy $\mathcal{H}(\mathbf{s}) = -\mathbb{E}_{\mathbf{s} \sim p(\mathbf{s})} \left[ \log p(\mathbf{s}) \right]$
5: // POLICY LEARNING
6: **for** each training step $t$ **do**
7:     // REWARD LEARNING
8:     **if** iteration $\% \ K == 0$ **then**
9:         Generate queries from replay buffer $\{(\sigma^0, \sigma^1)\}_{i=1}^{N_{\text{query}}} \sim \mathcal{B}$ and corresponding human feedback $(y_i)_{i=1}^{N_{\text{query}}}$
10:        Update reward model $\widehat{r}_\psi$ according to equation 2
11:        // RELABEL BOTH EXTRINSIC AND INTRINSIC REWARD
12:        Relabel entire replay buffer $\mathcal{B}$ using $\widehat{r}_\psi$
13:        Relabel extrinsic reward $r_\psi^{\text{ext}}(\mathbf{s}_j, \mathbf{a}_j) = \hat{r}_{\text{mean}}(\mathbf{s}_j, \mathbf{a}_j)$
14:        Relabel intrinsic reward $r_\psi^{\text{int}}(\mathbf{s}_j, \mathbf{a}_j) = \hat{r}_{\text{std}}(\mathbf{s}_j, \mathbf{a}_j)$
15:    **end if**
16:    // UPDATE POLICY
17:    **for** each gradient step **do**
18:        Sample minibatch from replay buffer $\{(\mathbf{s}_j, \mathbf{a}_j, \mathbf{s}_{j+1}, r_\psi^{\text{ext}}(\mathbf{s}_j, \mathbf{a}_j), r_\psi^{\text{int}}(\mathbf{s}_j, \mathbf{a}_j))\}_{j=1}^B \sim \mathcal{B}$
19:        Update $\beta_t \leftarrow \beta_0 (1 - \rho)^t$
20:        Let $r_j^{\text{total}} \leftarrow r_j^{\text{ext}} + \beta_t \cdot r_j^{\text{int}}$
21:        Update $\phi$ with transitions $\{(\mathbf{s}_j, \mathbf{a}_j, \mathbf{s}_{j+1}, r_\psi^{\text{total}}(\mathbf{s}_j, \mathbf{a}_j))\}_{j=1}^B$
22:    **end for**
23:    // INTERACTION WITH ENVIRONMENT
24:    **for** each timestep $t$ **do**
25:        Collect $\mathbf{s}_{t+1}$ by taking $\mathbf{a}_t \sim \pi_\phi(\mathbf{a}_t | \mathbf{s}_t)$
26:        Store transitions $\mathcal{B} \leftarrow \mathcal{B} \cup \{(\mathbf{s}_t, \mathbf{a}_t, \mathbf{s}_{t+1}, \widehat{r}_\psi(\mathbf{s}_t, \mathbf{a}_t))\}$
27:    **end for**
28: **end for**

---

| Hyperparameter | Value |
| --- | --- |
| Segment Length | 50 |
| Number of Unsupervised Pre-training Steps | 9000 |
| Interaction time | 10000 |
| Reward batch | 50 (Door Close, Door Open), 100 (Drawer Open) |
| Total feedback | 1000 (Door Close), 5000 (Door Open), 10000 (Drawer Open) |
| Sampling scheme | Disagreement Sampling |
| Reward model number of layers | 3 |
| Reward model number of hidden units | 256 |
| Reward model activation functions | LeakyRelu |
| Reward model output activation function | TanH |
| Number of reward functions | 3 |
| Optimizer | Adam (Kingma & Ba, 2015) |
| Initial learning rate | 0.0003 |

Table 2: Hyperparameters of the PEBBLE algorithm.

| Hyperparameter | Value | | Hyperparameter | Value |
|---|---|---|---|---|
| Initial temperature | 0.1 | | Hidden units per each layer | 256 |
| Learning rate | 0.0003 | | Batch Size | 512 |
| Critic target update freq | 2 | | Critic EMA $\tau$ | 0.005 |
| $(\beta_1, \beta_2)$ | $(.9, .999)$ | | Discount $\gamma$ | .99 |
| Optimizer | Adam (Kingma & Ba, 2015) | | | |

Table 3: Hyperparameters of the SAC algorithm. Most hyperparameters values are unchanged across environments with the exception for learning rate.

## B  EXPERIMENTAL DETAILS

### B.1  TRAINING DETAILS

For our method, we use the publicly released implementation repository of the PEBBLE algorithm (`https://github.com/pokaxpoka/B_Pref`) with the full list of hyperparameters specified in Table 2 and the SAC algorithm (`https://github.com/denisyarats/pytorch_sac`) with the full list of hyperparameters specified in Table 3.

### B.2  IMPLEMENTATION DETAILS OF EXPLORATION METHODS

For all exploration methods, we follow implementation of the publicly release repository of RE3 (`https://github.com/younggyoseo/RE3`) to update policy with $r^{\texttt{total}} = r^{\texttt{ext}} + \beta_t \cdot r^{\texttt{int}}$.

**RUNE.** We emphasize one simplicity in the implementation of RUNE. Following relabeling technique in Lee et al. (2021b), after each update in reward learning step, we relabel the entire replay buffer $B$ to store both predicted reward functions learned from human feedback (extrinsic rewards $r^{\texttt{ext}}$), and uncertainty in updated reward functions (intrinsic rewards $r^{\texttt{int}}$ in RUNE). This is because our learned reward functions parameters remain unchanged during RL training or agent interaction with the environment between subsequent feedback sessions. Thus, in RL training, $r^{\texttt{total}} = r^{\texttt{ext}} + \beta_t \cdot r^{\texttt{int}}$ can be directly taken from replay buffer. As for hyperparameter related to exploration, we consider $\beta_0 = 0.05$, $\rho \in \{0.001, 0.0001, 0.00001\}$.

**State Entropy Maximization. Seo et al. (2021b)** We use intrinsic reward $r^{\texttt{int}}(\mathbf{s}_t) = ||\mathbf{s}_t - \mathbf{s}_t^{k\text{-NN}}||_2$ followed Seo et al. (2021b) and Liu & Abbeel (2021). We use raw state space to compute entropy values as intrinsic rewards. We use code for unsupervised pre-training from the publicly release implementation repository of PEBBLE algorithm (`https://github.com/pokaxpoka/B_Pref`) to compute $k$-NN and estimate state entropy. As the size of replay buffer grows significantly as training time increases, we sample a random mini-batch $B'$ from replay buffer $B$ and compute $k$-NN with respect to random mini-batch. We use minibatch size 512, which is reported as a default hyperparameter in Seo et al. (2021b). As for hyperparameters related to exploration, we consider $\beta_0 = 0.05$, $\rho \in \{0.001, 0.0001, 0.00001\}$, and $k \in \{5, 10\}$.

**Disagreement. Pathak et al. (2019)** We train an ensemble of 5 forward dynamics model $g_i$, following original results in Pathak et al. (2019). These dynamics models are trained to predict next state $\mathbf{s}_{t+1}$ based on current state and action $(\mathbf{s}_t, \mathbf{a}_t)$, i.e. minimize $\sum_{i=1}^{5} ||g_i(\mathbf{s}_{t+1}|\mathbf{s}_t, \mathbf{a}_t) - \mathbf{s}_{t+1}||_2$. We use intrinsic reward proportional to variance in ensemble of predictions $r^{\texttt{int}}(\mathbf{s}_t, \mathbf{a}_t) = Var\{g_i(\mathbf{s}_{t+1}|\mathbf{s}_t, \mathbf{a}_t)\}_{i=1}^{N}$. For consistency we use same neural network architecture and optimization hyperparameters as reward functions specified in Table 2. We follow implementation of Disagreement agent in publicly released repository URLB (`https://anonymous.4open.science/r/urlb`). As for hyperparameter related to exploration, we consider $\beta_0 = 0.05$, $\rho \in \{0.001, 0.0001, 0.00001\}$.

**ICM. Pathak et al. (2017)** We train a single forward dynamics model $g$ to predict next state $\mathbf{s}_{t+1}$ based on current state and action $(\mathbf{s}_t, \mathbf{a}_t)$, i.e. minimize $||g(\mathbf{s}_{t+1}|\mathbf{s}_t, \mathbf{a}_t) - \mathbf{s}_{t+1}||_2$. We use intrinsic reward proportional to prediction error of trained dynamics model, i.e. $r^{\texttt{int}}(\mathbf{s}_t, \mathbf{a}_t) = ||g(\mathbf{s}_{t+1}|\mathbf{s}_t, \mathbf{a}_t) - \mathbf{s}_{t+1}||_2$. For consistency we use same neural network architecture and optimization hyperparameters as reward functions specified in Table 2. We follow implementation of ICM agent

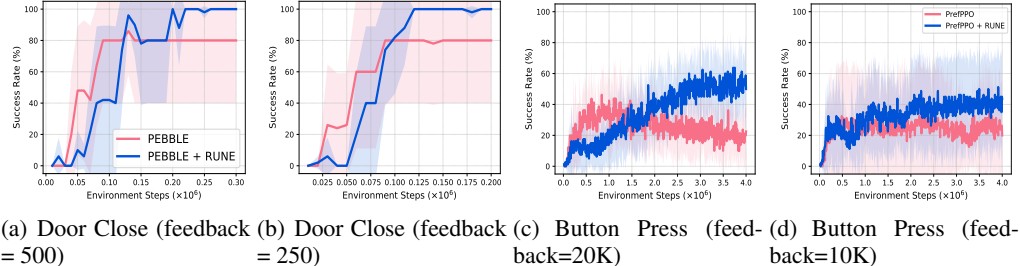

(a) Door Close (feedback = 500)  (b) Door Close (feedback = 250)  (c) Button Press (feedback=20K)  (d) Button Press (feedback=10K)

Figure 5: RUNE performs better than PEBBLE and PrefPPO in variety of manipulation tasks using different budgets of feedback queries. The solid/dashed line and shaded regions represent the mean and standard deviation, respectively, across five runs.

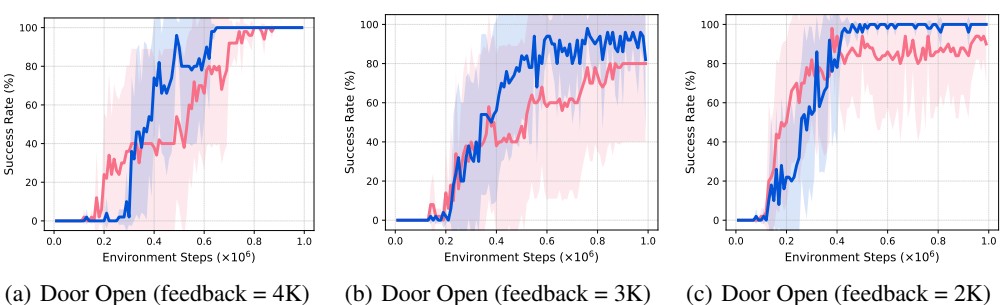

(a) Door Open (feedback = 4K)  (b) Door Open (feedback = 3K)  (c) Door Open (feedback = 2K)

Figure 6: RUNE performs better than PEBBLE in Door Open using different budgets of feedback queries. The solid/dashed line and shaded regions represent the mean and standard deviation, respectively, across five runs.

in publicly released repository URLB (https://anonymous.4open.science/r/urlb). As for hyperparameter related to exploration, we consider $\beta_0 = 0.05$, $\rho \in \{0.001, 0.0001, 0.00001\}$.

## C  ADDITIONAL EXPERIMENT RESULTS

**Improving Feedback-Efficiency.** We provide additional experimental results on various tasks from Meta-World benchmark Yu et al. (2020) using different budgets of feedback queries during training. We report asymptotic evaluation success rate at the end of training in Table 1. Here we report corresponding learning curves and additional results of wider choices of total feedback queries. We observe RUNE performs consistently at least same or better than PEBBLE baseline in different tasks with different budgets of feedback queries.

**Improving Sample-Efficiency.** As shown in Figure 9(a), RUNE outperforms other existing exploration baselines measured on the episode return of Pick Place. This shows applicability of our method on harder tasks.

**Reward Functions Learned by RUNE.** We provide additional analysis of reward functions learned by RUNE, compared to true reward function defined by Meta-World environments. Figure 10 demonstrates in all tasks we consider, learned reward functions optimized by RUNE is well-aligned to true reward from teacher feedback, because it can capture diverse patterns of true reward values from the environment.

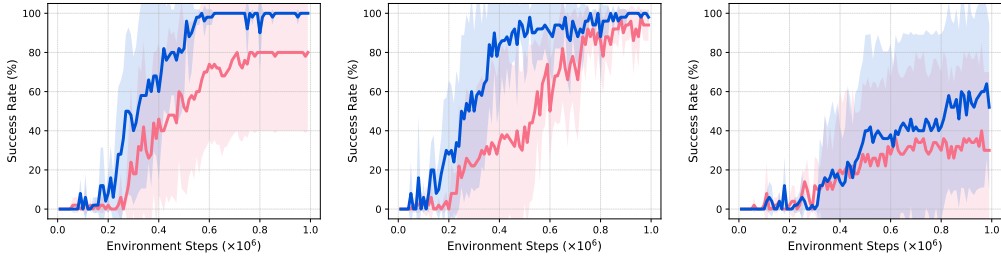

(a) Drawer Open (feedback = 7K)  (b) Drawer Open (feedback = 5K)  (c) Drawer Open (feedback = 3K)

Figure 7: RUNE performs better than PEBBLE in Drawer Open using different budgets of feedback queries. The solid/dashed line and shaded regions represent the mean and standard deviation, respectively, across five runs.

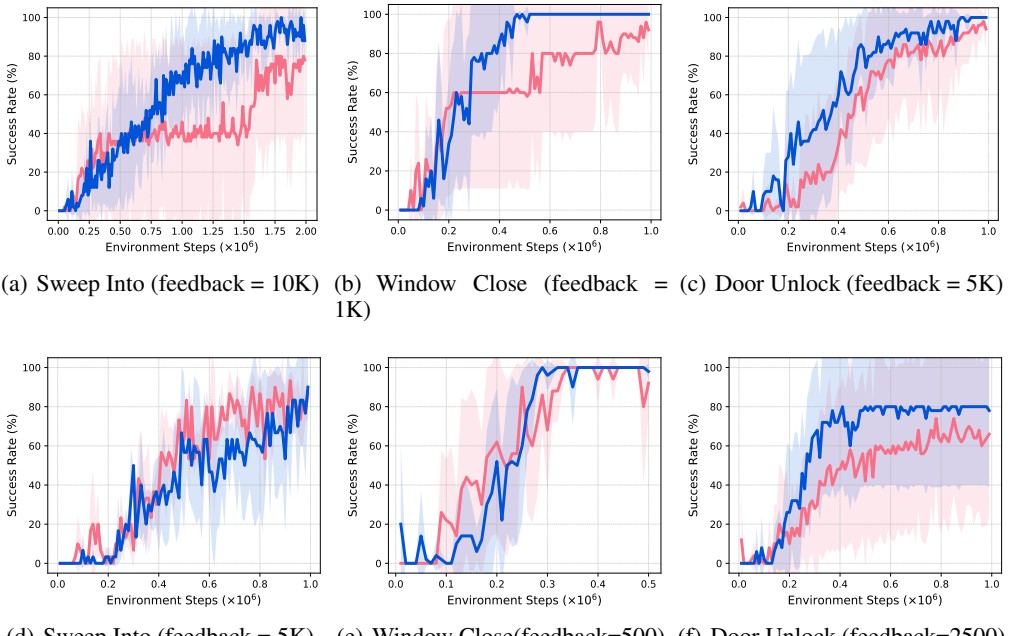

(a) Sweep Into (feedback = 10K)  (b) Window Close (feedback = 1K)  (c) Door Unlock (feedback = 5K)

(d) Sweep Into (feedback = 5K)  (e) Window Close(feedback=500)  (f) Door Unlock (feedback=2500)

Figure 8: RUNE performs better than PEBBLE in variety of manipulation tasks using different budgets of feedback queries. The solid/dashed line and shaded regions represent the mean and standard deviation, respectively, across five runs.

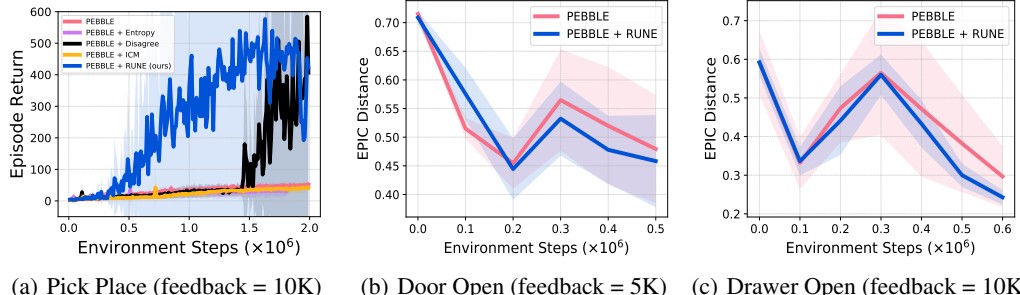

(a) Pick Place (feedback = 10K)  (b) Door Open (feedback = 5K)  (c) Drawer Open (feedback = 10K)

Figure 9: (a) RUNE achieves better sample-efficiency on manipulation tasks from Meta-World benchmark. (b/c) EPIC distance between the ensemble of learned reward functions and true reward in task environments. The solid/dashed line and shaded regions represent the mean and standard deviation, respectively, across five runs.

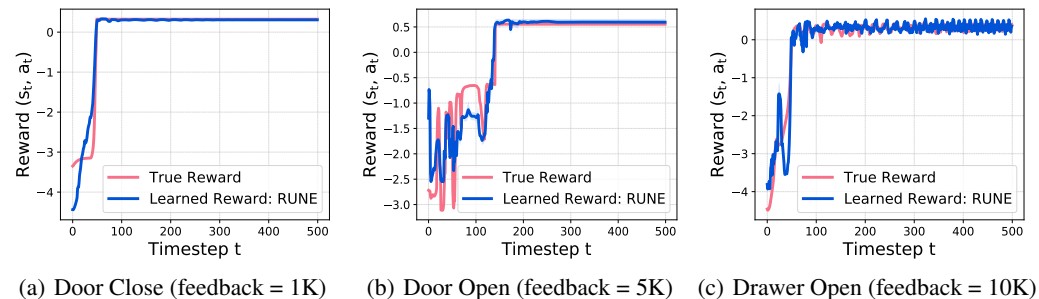

(a) Door Close (feedback = 1K)  (b) Door Open (feedback = 5K)  (c) Drawer Open (feedback = 10K)

Figure 10: Time series of normalized learned reward (blue) and the ground truth reward (red) using rollouts from a policy optimized by RUNE.

