# OpenReview forum: "Reward Uncertainty for Exploration in Preference-based Reinforcement Learning"
_ICLR.cc/2022/Conference — ICLR 2022 Poster_

### Official Review · Reviewer_v2Tv · 2021-10-29

**Correctness:** 3
**Technical Novelty And Significance:** 2
**Empirical Novelty And Significance:** 3
**Recommendation:** 6
**Confidence:** 3

**Main Review:**

The paper is well written and clear to understand. The method can be considered sound, as it is based on established building-blocks. Some minor issues should be improved, but have no substantial impact:
- Ensemble disagreement (mentioned in abstract) and ensemble variance are not the same thing
- Preference notation is introduced as (1,0),(0,1), but seq_0 > seq_1 is used
- Indifference seems to be not handled, but is introduced. (Which is ok, but should be clarified)
- The agent is not encouraged to visit more "unseen" states, but "uncertain" states (wrt. the true reward)

Additionally, the method has a hidden assumption, which is quite common, but should be made explicit: The preference feedback is assumed to be stationary and acyclic. Without these assumptions, the surrogate reward may not converge towards a low variance solution and the mean reward can not sufficiently reflect the (unknown, human) preference function. In the cyclic preference, case this not possible at all (with a scalar reward) and in the non-stationary case, the (unweighted) mean is not a good estimator. Both assumptions are often violated by real-world human preference feedback.

The evaluation is ok, but more domains should be added for allowing to derive more general conclusions. The ablation study is a good addition, with the exception of the "Quality of learned reward functions" section. As acknowledged by the authors, surrogate rewards may have a different scale than the underlying, true rewards (also when not using tanh). Furthermore, even rewards that differ not only in scale, can induce the same (optimal) policy. Therefore, the comparison should be done in terms of a different metric, like ranking error over states and/or sequences. Even rescaling the learned reward would help, because with the current plot, alignment is not clearly visible.

However, surrogate-reward uncertainty-based exploration is not new and the according advantages have also been shown before (e.g. Akrour 2011 - for more, see "Trajectory Generation" in https://jmlr.org/papers/volume18/16-634/16-634.pdf). None of these methods have been applied to PEEBLE/DeepRL (as far as the reviewer is aware), but this still reduces the validity of the claims. Furthermore, the idea of using ensemble variance as intrinsic reward bonus is also already known from plain RL (as mentioned by the authors). Resultingly, the publication is applying known methods to a different framework (PEEBLE). The impact of this issue can maybe reduced by clearly stating the differences to other preference-based exploration strategies.

However, even if these issues are resolved, the degree of novelty will likely stay limited. Therefore, the evaluation should be substantially improved (besides adding more domains, potentially adding some results from the appendix). Some suggestions:
- Effect of different beta values
- Comparison to "comparable" formulations, for intrinsic reward, based on preference uncertainty (e.g. ensemble disagreement, instead of variance)
- Optimal/terminal policy in terms of preference queries (instead of only two query counts)

**Summary Of The Paper:**

The authors introduce a method for integrating reward uncertainty into the exploration/exploitation tradeoff in a preference-based reinforcement learning setting. The method uses a surrogate reward ensemble variance as intrinsic reward bonus.

**Summary Of The Review:**

The proposed method is of interest within the domain of preference-based reinforcement learning, as guiding exploration can reduce human operator workload. The novelty of the approach is limited and the claims have been shown before, but in a different framework. This limitation has to be overcome or leveled by a thorough evaluation, allowing generalized conclusions.

---

> ### Author Response · Authors · 2021-11-20
> **Authors' Response (Part 2/2)**
>
> (Continue)
>
> ---
> **Q3.** Confusion in quality of learned reward function
>
> **A3.** Thanks for pointing out deficiencies in our figure. We update the figure about Quality of learned reward functions in Figure 6 of Appendix. To better show the alignment between learned reward and true reward, we normalize both values with respect to their means and standard deviations.
>
> Regarding confusion and inconsistency in preference notation, we update descriptions in Section 3.
>
> ---
> **Q4.** Assumption in preference feedback
>
> **A4.** Thanks for raising an important assumption in preference feedback used for training. For all experiments, we use oracle scripted teacher preferences [5]: Given a pair of behaviors $(\sigma^0, \sigma^1)$, the scripted teacher always provides a clean preference based on the sum of underlying true task reward values for each segment. We hope this clarifies our experiment setup and addresses concerns of unstable or cyclical feedback.
>
> ---
> [1] Burda, Yuri, et al. "Exploration by random network distillation." arXiv preprint arXiv:1810.12894 (2018).
>
> [2] Pathak, Deepak, Dhiraj Gandhi, and Abhinav Gupta. "Self-supervised exploration via disagreement." International conference on machine learning. PMLR, 2019.
>
> [3] Zhang, Yunzhi, Pieter Abbeel, and Lerrel Pinto. "Automatic curriculum learning through value disagreement." Advances in Neural Information Processing Systems 33 (2020).
>
> [4] Lee, Kimin, et al. "Sunrise: A simple unified framework for ensemble learning in deep reinforcement learning." International Conference on Machine Learning. PMLR, 2021.
>
> [5] Lee, Kimin, et al. "B-Pref: Benchmarking Preference-Based Reinforcement Learning." (2021).

---

> > ### Comment · Reviewer_v2Tv · 2021-11-22
> > **Reviewers' Response**
> >
> > The introduced changes resolve several of my complaints. Especially the substantially improved result section is a great addition. Novelty is still limited and distinction from existing approaches could still be improved. Especially wrt. the very strong claim "For the first time, we show that exploration can improve the sample- and feedback-efficiency of preference-based RL algorithms." However, the remaining advantages still qualify the paper for a substantial score increase.

---

> > > ### Author Response · Authors · 2021-11-22
> > > **Thank you for your additional comments**
> > >
> > > We are happy to hear that our rebuttal addressed your concerns well.
> > >
> > > We will further improve the distinction from other works based on your comments.
> > >
> > > Thank you again for the valuable suggestions and comments to add, which we will incorporate in the revision to strengthen our paper.
> > >
> > > If you have any remaining suggestions or concerns, please let us know!
> > >
> > > Best, Authors.

---

> ### Author Response · Authors · 2021-11-20
> **Authors' Response (Part 1/2)**
>
> We sincerely thank you for your helpful feedback and insightful comments. We appreciate that our paper is recognized for several positive aspects: (1) Tackle an underexplored problem setting, (2) our proposed method makes intuitive sense, is simple, scalable, and easy to implement, (3) writing and structure of paper is clear. We address your comments and questions below:
>
> ---
> **Q1.** Limited experimental evaluations of algorithm
>
> **A1.** To address concerns on performance gains, we added additional experiment results (Table 1) in Section 5.3 of the revised draft. We run experiments on 6 tasks in the Meta-World benchmark. For each task, we consider different budgets of feedback queries during training to compare the feedback efficiency PEBBLE and RUNE. We use asymptotic success rate at the end of training as the evaluation metric. We also report corresponding learning curves in Figure 7. For each experiment, we report mean and standard deviations across 5 runs.
>
> From both asymptotic success rate and sample efficiency, RUNE shows consistently improvements than PEBBLE baseline in various tasks. This shows the benefit of efficient exploration in preference-based RL algorithms in order to improve both sample-efficiency and feedback-efficiency. Here, we remark that exploration methods have more significant performance gain in harder environments,  where performance of naive preference-based RL algorithms (e.g. PEBBLE) is not good enough.
>
> Additionally, we provide additional ablation study in Section 5.4, we compare performance of PEBBLE and RUNE under different numbers of queries available per feedback session. We show robustness in performance of RUNE than PEBBLE with respect to the number of available queries per feedback session. We hope these additional experiment results address your concerns and add validity to our proposed approach.
>
> ---
> **Q2.** Novelty of our proposed approach - Differences from other preference-based exploration
>
> **A2.** We agree that uncertainty driven exploration methods have been proposed and studied previously in other RL algorithms. Previous uncertainty driven exploration methods capture epistemic uncertainty in state visitations [1], transition dynamics [2], Q-functions [3] or value functions [4]. However, we highlight our contribution as the first attempt to extract uncertainty from human feedback through learned reward functions.
>
> We emphasize that our motivation of RUNE, i.e. epistemic uncertainty over learned reward functions, is to develop human-guided exploration in preference-based RL. The ensemble of reward functions learned from human feedback is the only medium that communicates between RL agents and human teachers, and delivers information from human feedback. As we expect RL agents to achieve behaviors well-aligned with human intents, efficiently extracting useful information from human instructions is important to drive agent learning. RUNE encourages visitations of states and actions more uncertain with respect to human feedback, which provides a different perspective towards exploration in RL from uncertainty in state visitations or transition dynamics. Indeed, in our experiments, we found that capturing uncertainty of human feedback is more effective than other existing exploration methods. We hope this clarifies motivation and novelty in our approach and distinction from previous exploration methods.
>
> In addition, we provide an explanation about why naive exploration based on novelty in state visitations isn’t enough in preference-based RL. After states and actions are fully visited, there can still be high variance in predicted reward values of frequently visited states and actions because it receives fewer feedback queries. Such uncertainty in learned rewards may reflect instability in predicted reward values for RL policy training. We propose to explore these uncertainty states and actions with respect to learned rewards to showcase more these behaviors in environment interactions and in subsequent feedback sessions. This can improve reward learning as human preferences are tailored to reduce these uncertainties. We hope this explanation clarifies why exploration based on novelty in state visitations isn’t enough.
>
> For more detailed comparisons with existing exploration schemes specifically applied to preference-based RL setting, we updated Section 2 with a more comprehensive review of related works.
>
> ---
> (Citations included in Part 2)

---

### Official Review · Reviewer_WzMo · 2021-11-02

**Correctness:** 2
**Technical Novelty And Significance:** 3
**Empirical Novelty And Significance:** 3
**Recommendation:** 6
**Confidence:** 4

**Main Review:**

Exploration strategies for preference-based RL is underexplored and this paper takes a step towards practical RL methods that can learn with humans-in-the-loop. Although the idea is not groundbreaking given existing work, RUNE is simple, scalable, and easy to implement. Adding an uncertainty-based intrinsic reward to guide preference-based RL is intuitive and at its core, I think the underlying idea can improve existing methods.

However, I do have concerns about this particular instantiation of the approach. First, the method can be better explained and it's unclear how RUNE is trained:

- Is each model in the ensemble trained using the same set of queries and feedback? Are the models trained independently given their respective data?
- In general, there could be multiple reward functions that give rise to the same preference feedback (e.g., a translation of the reward function). In this case, each model in the ensemble might learn correct but different reward functions, yet the standard deviation of the state rewards would be non-zero? Is this still a meaningful estimate of uncertainty?
- In the above case, is the average reward indicative of the actual extrinsic reward?

Second, the main claims are not well-validated by the experiments. While I appreciate the authors used rather complex robot domains, it is difficult to understand if RUNE actually increases exploration:

- Iit is unclear if the better performance is due to extra exploration or simply the use of a better reward estimator, i.e., the ensemble. I encourage the authors to quantitatively compare the amount of exploration when RUNE is used with PEBBLE. If the current domains are too complex, a simple gridworld-type domain (or small-scale continuous domain) can be used.
- If my interpretation is correct, Fig 5b seems to indicate that less exploration is performed with RUNE. We would expect that disagreement to be higher at the middle portions as the method explores uncertain states?
- For the experiments, how many runs are performed for each method? Is there noise in the simulation human feedback? Does the simulated human prefer one trajectory above all others or is there some ambiguity? These details are important to ascertain the robustness of the method and results. Also, please use a different color for the methods in Fig 4; I am unable to compare the standard deviations.
- The analysis on the quality of learned rewards can be improved. Normalizing the rewards to a standard range, e.g., [0,1], can aid comparisons in Fig 5.c. Again, using a gridworld here may help to better illustrate that a reasonable reward function is learnt. At present, the figure shows the rewards from a single trajectory rather than the state space.




**Summary Of The Paper:**

This work proposes an ensemble-based intrinsic reward to improve exploration in preference-based RL. The key idea is to incorporate uncertainty in teacher preferences as an intrinsic reward. An ensemble of reward functions is used to capture this uncertainty. The paper discusses experiments on three robot tasks with ablation studies to investigate the method's performance.

**Summary Of The Review:**

Exploration for preference-based RL is an interesting problem, but the current paper has multiple issues that should be addressed. I hope the authors can provide some clarification my questions above.

---

> ### Author Response · Authors · 2021-11-20
> **Authors' Response (Part 2/2)**
>
> (Continue)
>
> ---
> **Q4.**  Compare disagreement in selected queries of PEBBLE and RUNE
>
> **A4.** Disagreement with respect to a pair of segment $(\sigma^0, \sigma^1)$ under an ensemble of reward functions $ \hat{r}_{i=1}^{N}$ is variance among $N$ preference predictions. Intuitively, higher disagreement indicates the reward function is uncertain to recognize a preference between a pair of behaviors. Such property is undesired in learned reward functions. However, reward functions learned from RUNE show higher confidence in preference prediction. This therefore indicates improved confidence in reward learning.
>
> ---
> **Q5.**  Robustness of proposed method and experiment results
>
> **A5.** Thank you for questions about detailed experiment setups. We add additional clarifications below:
>
> For all experiments, we report mean and standard deviation across 5 runs, respectively. We use oracle scripted teacher preferences [1]: Given a pair of behaviors $(\sigma^0, \sigma^1)$, the scripted teacher always provides a clean preference based on the sum of underlying true reward values for each segment. While adding noises in simulated human preferences is helpful to analyze robustness of preference-based RL algorithms, we leave it as an interesting future direction for further investigations.
>
> Sorry for the inconvenience in visualization in original Figure 4. To address concerns on performance gains, we added additional experiment results (Table 1) in Section 5.3 of the revised draft. We run experiments on 6 tasks in the Meta-World benchmark. For each task, we consider different budgets of feedback queries during training to compare the feedback efficiency PEBBLE and RUNE. We use asymptotic success rate at the end of training as the evaluation metric. We also report corresponding learning curves in Figure 7. For each experiment, we report mean and standard deviations across 5 runs.
>
> From both asymptotic success rate and sample efficiency, RUNE shows consistently improvements than PEBBLE baseline in various tasks. This shows the benefit of efficient exploration in preference-based RL algorithms in order to improve both sample-efficiency and feedback-efficiency. Here, we remark that exploration methods have more significant performance gain in harder environments,  where performance of naive preference-based RL algorithms (e.g. PEBBLE) is not good enough.
>
> Additionally, we provide additional ablation study in Section 5.4, we compare performance of PEBBLE and RUNE under different numbers of queries available per feedback session. We show robustness in performance of RUNE than PEBBLE with respect to the number of available queries per feedback session. We hope these additional experiment results address your concerns and add validity to our proposed approach.
>
> ---
> [1] Lee, Kimin, et al. "B-Pref: Benchmarking Preference-Based Reinforcement Learning." (2021).

---

> > ### Comment · Reviewer_WzMo · 2021-11-23
> > **Thanks for the response.**
> >
> > Thanks for your detailed response and for clarifying your experimental details, especially on the use of ensembles for all methods. I'm positive about the overall direction of this work and RUNE as a whole. The added experiments on the budget-limited scenarios also shows the method has promise.
> >
> > There are aspects of the paper that could be improved (e.g., experiments showing robustness to label noise, theoretical results). The issue regarding multiple reward functions by the ensemble also remains unclear; possibly normalizing to $(-1,1)$ addresses some of this but is by no means a solution to my knowledge. Given this issue, it is surprising to me that RUNE works as well as it does. What RUNE seems to provide is *better* rather than *more* exploration and the paper would benefit from a thorough discussion how the trajectories are better. Despite these issues, I think the paper would generate some discussion and would be an interesting addition to conference.

---

> > > ### Author Response · Authors · 2021-11-24
> > > **Thank you for additional comments and response about reward learning issue**
> > >
> > > We are happy to hear that our clarifications and additional experiments address your concerns well. We appreciate your overall positive attitude towards our work. Thank you for valuable suggestions and constructive feedback (e.g. experimental results to show robustness to label noise, theoretical justifications, and evaluations of better trajectories), which we will incorporate in the revision to strengthen our paper.
> > >
> > > We address your questions and concerns about learning multiple reward functions at different scales below:
> > >
> > > We want to further clarify that in all experiments, we independently normalize each reward function predictors to the range of $(-1, 1)$ to reduce issues related to learning multiple reward functions at different scales. Instead of first taking the average value of predicted rewards by the ensemble and then normalizing these mean values, we independently normalize each reward function in the ensemble to the same scale. We believe that this makes standard deviations of predicted rewards meaningful and our proposed exploration method RUNE reasonable.
> > >
> > > To further demonstrate why this is not a weakness of our work, an ensemble of independently normalized reward predictors has been widely used and studied in previous preference-based RL literatures [1][2]. Similar types of concerns can be handled by various techniques: Christiano, et al. normalized by measuring running mean and standard deviation of rewards [1]; Lee, et al. normalized by using tanh activation function in the output layer of neural network architecture. Here we follow normalization used by Lee, et al. [2]. One may argue that counterexamples can always happen, but in practice independent normalization works pretty well as a reasonable solution. We hope these in-depth discussions and popular usages in previous works add validity to our approach.
> > >
> > > We hope our additional response clarifies your questions about our approach. If you have additional concerns or comments, please let us know.
> > >
> > > ---
> > >
> > > [1] Christiano, Paul, et al. "Deep reinforcement learning from human preferences." arXiv preprint arXiv:1706.03741 (2017).
> > >
> > > [2] Lee, Kimin, Laura Smith, and Pieter Abbeel. "PEBBLE: Feedback-Efficient Interactive Reinforcement Learning via Relabeling Experience and Unsupervised Pre-training." arXiv preprint arXiv:2106.05091 (2021).

---

> ### Author Response · Authors · 2021-11-20
> **Authors' Response (Part 1/2)**
>
> We sincerely thank you for your helpful feedback and insightful comments. We appreciate that our paper is recognized for several positive aspects: (1) Tackle an underexplored problem setting, (2) our proposed method makes intuitive sense, is simple, scalable, and easy to implement, (3) writing and structure of paper is clear. We address your comments and questions below:
>
> ---
> **Q1.**  How ensemble of reward functions is trained
>
> **A1.** We use different random initialization for each model in the ensemble. We use the same set of training data, i.e. set of feedback queries, but different random batches to train each model in the ensemble. Parameters of each model are independently optimized to minimize the cross entropy loss of their respective batch of training data.
>
> ---
> **Q2.**  Reward learning
>
> **A2.** We appreciate you raising such concern in reward learning from human preferences. We agree that there can be multiple reward functions that align to the same set of preferences. Such differences in scale may lead to unreliable RL training. However, to address this issue, we bound all predicted reward values to a normalized range of $(-1, 1)$ by adding tanh activation function after the output layer in all experiments. We believe this can make learned reward functions in the ensemble more consistent, and make our proposed estimate of uncertainty, i.e. standard deviations between predicted values, meaningful.
>
> ---
> **Q3.**  Source of better performance
>
> **A3.** Better performance is from extra exploration not from reward ensembles, as we fix all other experiment setups completely the same, except the choice of intrinsic reward for exploration. All baselines use an ensemble of 3 reward functions from human preferences. In other words , the only difference between different learning curves is the choice of exploration methods, i.e. no exploration, RUNE, StateEnt [1], ICM [2], or Disagreement [3]. We hope this clarifies your concerns about the source of better performance.
>
> ---
> [1] Liu, Hao, and Pieter Abbeel. "Behavior from the void: Unsupervised active pre-training." arXiv preprint arXiv:2103.04551 (2021).
>
> [2] Pathak, Deepak, et al. "Curiosity-driven exploration by self-supervised prediction." International conference on machine learning. PMLR, 2017.
>
> [3] Pathak, Deepak, Dhiraj Gandhi, and Abhinav Gupta. "Self-supervised exploration via disagreement." International conference on machine learning. PMLR, 2019.

---

> ### Author Response · Authors · 2021-11-22
> **A gentle reminder**
>
> Dear Reviewer WzMo,
>
> Thank you for your time and efforts in reviewing our paper.
>
> We kindly remind that the discussion period will end soon.
>
> We sincerely hope that our response and results of the supporting experiments successfully clarify your concerns.
>
> We just wonder whether we could have the last chance to address your further concerns or questions (if you have any).
>
> Thank you very much!
>
> Authors

---

### Official Review · Reviewer_VUcZ · 2021-11-04

**Correctness:** 3
**Technical Novelty And Significance:** 2
**Empirical Novelty And Significance:** 2
**Recommendation:** 6
**Confidence:** 4

**Main Review:**

strengths: A simple solution to the problem of epistemic uncertainty driven exploration for RL from preferences. The experimental results demonstrate improved sample efficiency over other baselines. Also, the writing and the structure of the paper is clear.

weaknesses: The paper suggests that capturing the epistemic uncertainty at the reward prediction level is better than capturing it at the level of the dynamics. Although I find this intuition correct, you could show this theoretically as well since the reward uncertainty incorporates the state transition uncertainty as well. I'd like to see some theoretical analysis that would support the experiments. Also, I see this paper as a direct application of the epistemic uncertainty driven exploration in RL from preferences that make its novelty small.

C1: "We take advantage of an ensemble of reward models ... which is not available in other traditional RL settings" - What is the challenge here? To have a reward model or an ensemble of reward models? I don't think that any of those is challenging. The disagreement method doesn't need to happen over the forward dynamics model but can also happen over the rewards. In fact, the intuition here is that you model the epistemic uncertainty of the predictor of interest. If you want to cover more states you model the epistemic uncertainty over the forward states, if you want to cover more downstream task-related states you model the epistemic uncertainty over the reward model.

C2: "Quality of learned reward functions" - The plot doesn't seem to add anything valuable. The scales are different (you explain why) but that could also have been the result of learning a wrong reward function. What you want perhaps to show instead is that for sampled $s_t, a_t$ pairs the predicted reward and the true reward is correlated.

**Summary Of The Paper:**

With this work, the authors propose a bayesian active learning approach to the problem of reinforcement learning from preferences. To do this, they model the epistemic uncertainty of the reward function to intrinsically motivate the RL agents to explore. The solution demonstrates improved sample efficiency.

**Summary Of The Review:**

I find the paper clearly written and a direct application of epistemic uncertainty driven exploration in RL from preferences. Although simple, its novelty is marginal. Also, it lacks theoretical justification (e.g. bayesian RL intuitions, why uncertainty over the rewards might work better than uncertainty over dynamics, aleatoric vs epistemic uncertainty, etc.). Finally, the experiments, although they support the argument, there are experiments that don't contribute to the discussion (C2)

---

> ### Author Response · Authors · 2021-11-20
> **Authors' Response (Part 2/2)**
>
> (Continue)
>
> **Q4.** Challenge in preference-based RL setting
>
> **A4.** Traditional RL setting assumes the existence of a good reward function, which can fully specify our desired task. However, one can expect that specifying such a reward function is difficult and time-consuming. As an alternative, preference-based RL algorithms do not require a predefined reward function. We learn appropriate reward function(s) from human feedback to deliver task objectives to RL agents. This is challenging because we can only use a limited number of feedback queries. To handle this issue, we introduce exploration as an intrinsic reward to speed up reward learning and policy training. Our experimental results show that exploration is indeed helpful under more complex tasks, in which case naive reward learning and policy training using preference-based RL algorithms are inefficient.
>
> ---
> **Q5.** Quality of learned reward function
>
> **A5.** Thanks for pointing out deficiencies in our figure. We update the figure about Quality of learned reward functions in Figure 6 of Appendix. To better show the alignment between learned reward and true reward, we normalize both values with respect to their means and standard deviations.

---

> > ### Comment · Reviewer_VUcZ · 2021-11-28
> > **thank you for addressing my comments**
> >
> > regarding Q1 - I appreciate the effort of the authors to run more baselines (a common request among other reviewers as well) and am happy to see that the gains are consistent. I've updated my score to reflect this.
> >
> > Q3, Q4 - I understand the motivation of this work I was just commenting on, what seems to me, a low hanging fruit. Perhaps my comment stems from the surprise that nobody has tried that before but on the other hand this is based on PEBBLE which is very recently got published.
> >
> > I'm happy with Q5 - the normalized versions seem to be matching the learned and the true reward.
> >
> > A few more suggestions:
> > - "The solid line and shaded regions represent the mean and standard deviation" - I think here you might need better to show the standard error instead of standard deviation.
> > - Also, Since you might have some extra space left, why don't you surface all the meta-world results in the main text?

---

> > > ### Author Response · Authors · 2021-11-29
> > > **Thank you for additional comments**
> > >
> > > We are happy to hear that our rebuttal response, additional experiments, and improvements in figures addressed your concerns well.
> > >
> > > We will further improve visualizations of learning curves and structures of presenting results based on your suggestions.
> > >
> > > Thank you again for the valuable suggestions and comments to add, which we will incorporate in the revision to strengthen our paper.
> > >
> > > If you have any remaining suggestions or concerns, please let us know!
> > >
> > > Best,
> > > Authors.

---

> ### Author Response · Authors · 2021-11-20
> **Authors' Response (Part 1/2)**
>
> We sincerely thank you for your helpful feedback and insightful comments. We appreciate that our paper is recognized for several positive aspects: (1) Tackle an underexplored problem setting, (2) our proposed method makes intuitive sense, is simple, scalable, and easy to implement, (3) writing and structure of paper is clear. We address your comments and questions below:
>
> ---
> **Q1.** Limited experimental evaluations of algorithm
>
> **A1.** To address concerns on performance gains, we added additional experiment results (Table 1) in Section 5.3 of the revised draft. We run experiments on 6 tasks in the Meta-World benchmark. For each task, we consider different budgets of feedback queries during training to compare the feedback efficiency PEBBLE and RUNE. We use asymptotic success rate at the end of training as the evaluation metric. We also report corresponding learning curves in Figure 7. For each experiment, we report mean and standard deviations across 5 runs.
>
> From both asymptotic success rate and sample efficiency, RUNE shows consistently improvements than PEBBLE baseline in various tasks. This shows the benefit of efficient exploration in preference-based RL algorithms in order to improve both sample-efficiency and feedback-efficiency. Here, we remark that exploration methods have more significant performance gain in harder environments,  where performance of naive preference-based RL algorithms (e.g. PEBBLE) is not good enough.
>
> Additionally, we provide additional ablation study in Section 5.4, we compare performance of PEBBLE and RUNE under different numbers of queries available per feedback session. We show robustness in performance of RUNE than PEBBLE with respect to the number of available queries per feedback session. We hope these additional experiment results address your concerns and add validity to our proposed approach.
>
> ---
> **Q2.** Theoretical justification
>
> **A2.**  We agree that theoretical proofs to support our intuition are interesting, but we would like to leave it as a future investigation.
>
> As an alternative, we provide an explanation about why naive exploration based on novelty in state visitations isn’t enough in preference-based RL. After states and actions are fully visited, there can still be high variance in predicted reward values of frequently visited states and actions because it receives fewer feedback queries. Such uncertainty in learned rewards may reflect instability in predicted reward values for RL policy training. We propose to explore these uncertainty states and actions with respect to learned rewards to showcase more these behaviors in environment interactions and in subsequent feedback sessions. This can improve reward learning as human preferences are tailored to reduce these uncertainties. We hope this explanation clarifies why exploration based on novelty in state visitations isn’t enough.
>
> ---
> **Q3.** Novelty of proposed approach
>
> **A3.** We agree that uncertainty driven exploration methods have been proposed and studied previously in other RL algorithms. Previous uncertainty driven exploration methods capture epistemic uncertainty in state visitations [1], transition dynamics [2], Q-functions [3] or value functions [4]. However, we highlight our contribution as the first attempt to extract uncertainty from human feedback through learned reward functions.
>
> We emphasize that our motivation of RUNE, i.e. epistemic uncertainty over learned reward functions, is to develop human-guided exploration in preference-based RL. The ensemble of reward functions learned from human feedback is the only medium that communicates between RL agents and human teachers, and delivers information from human feedback. As we expect RL agents to achieve behaviors well-aligned with human intents, efficiently extracting useful information from human instructions is important to drive agent learning. RUNE encourages visitations of states and actions more uncertain with respect to human feedback, which provides a different perspective towards exploration in RL from uncertainty in state visitations or transition dynamics. Indeed, in our experiments, we found that capturing uncertainty of human feedback is more effective than other existing exploration methods. We hope this clarifies motivation and novelty in our approach and distinction from previous exploration methods.
>
> ---
> [1] Burda, Yuri, et al. "Exploration by random network distillation." arXiv preprint arXiv:1810.12894 (2018).
>
> [2] Pathak, Deepak, Dhiraj Gandhi, and Abhinav Gupta. "Self-supervised exploration via disagreement." International conference on machine learning. PMLR, 2019.
>
> [3] Zhang, Yunzhi, Pieter Abbeel, and Lerrel Pinto. "Automatic curriculum learning through value disagreement." Advances in Neural Information Processing Systems 33 (2020).
>
> [4] Lee, Kimin, et al. "Sunrise: A simple unified framework for ensemble learning in deep reinforcement learning." International Conference on Machine Learning. PMLR, 2021.

---

> ### Author Response · Authors · 2021-11-22
> **A gentle reminder**
>
> Dear Reviewer VUcZ,
>
> Thank you for your time and efforts in reviewing our paper.
>
> We kindly remind that the discussion period will end soon.
>
> We sincerely hope that our response and results of the supporting experiments successfully clarify your concerns.
>
> We just wonder whether we could have the last chance to address your further concerns or questions (if you have any).
>
> Thank you very much!
>
> Authors

---

### Official Review · Reviewer_x6bk · 2021-11-05

**Correctness:** 3
**Technical Novelty And Significance:** 2
**Empirical Novelty And Significance:** 2
**Recommendation:** 5
**Confidence:** 4

**Main Review:**

The idea of efficient exploration is very important for reinforcement learning, especially for cases where human feedback is used. The authors propose an interesting approach of looking at reward model ensemble. At a high level, this idea makes sense, but looking at the evaluation, the effect of the algorithm on the results is very minor. In many of the plots, it is not easy to see a big difference between different methods. In addition, the evalutation is very limited. The authors use one problem (manipulation) and 3 tasks from that problem.

The paper is well written, but same citations are repeated over and over. This is both unnecessary and distrupts reading flow.

**Summary Of The Paper:**

The paper proposes an exploration method for "preference based Reinforcement Learning" methods, where human feedback is incorporated to the training regime. The authors use an ensemle of learned reward models and add an intrinsic reward based on disagreement (or uncertainity). The authors test the idea on robotic manipulation tasks from meta-world. The agent learns purely based on the feedback from a teacher that provides preference of trajectory one over another. The tasks presented are "door close", "door open" and "drawer open".
The authors combine their exploration strategy (RUNE) with the preference based learning method PEBBLE and compare it with PEBBLE with some other exploration strategies. The results show that the proposed method provide some improvement over others. The authors also compare with PEBBLE by using 700 feedback instead of 1000. The results show minor improvement.

**Summary Of The Review:**

The paper tackles an interesting problem for RL, the proposed method is incremental but partially novel. On the other hand, the evaluation is limited and the results are not significant.

---

> ### Author Response · Authors · 2021-11-20
> **Authors' Response**
>
> We sincerely thank you for your helpful feedback and insightful comments. We appreciate that our paper is recognized for several positive aspects: (1) Tackle an underexplored and important problem setting, (2) our proposed method makes intuitive sense, is simple, scalable, and easy to implement, (3) structure of paper is clear. We address your comments and questions below:
>
> ---
> **Q1.** Limited experimental evaluations of algorithm
>
> **A1.** To address concerns on performance gains, we added additional experiment results (Table 1) in Section 5.3 of the revised draft. We run experiments on 6 tasks in the Meta-World benchmark. For each task, we consider different budgets of feedback queries during training to compare the feedback efficiency PEBBLE and RUNE. We use asymptotic success rate at the end of training as the evaluation metric. We also report corresponding learning curves in Figure 7. For each experiment, we report mean and standard deviations across 5 runs.
>
> From both asymptotic success rate and sample efficiency, RUNE shows consistently improvements than PEBBLE baseline in various tasks. This shows the benefit of efficient exploration in preference-based RL algorithms in order to improve both sample-efficiency and feedback-efficiency. Here, we remark that exploration methods have more significant performance gain in harder environments,  where performance of naive preference-based RL algorithms (e.g. PEBBLE) is not good enough.
>
> Additionally, we provide additional ablation study in Section 5.4, we compare performance of PEBBLE and RUNE under different numbers of queries available per feedback session. We show robustness in performance of RUNE than PEBBLE with respect to the number of available queries per feedback session. We hope these additional experiment results address your concerns and add validity to our proposed approach.
>
> ---
> **Q2.** Inappropriate citations in paper-writing
>
> **A2.**  Thank you for your pointers. We removed repeated citations and improved the quality of writing in the revised draft.
>
> ---
> **Q3.** Novelty of proposed approach
>
> **A3.** We agree that uncertainty driven exploration methods have been proposed and studied previously in other RL algorithms. Previous uncertainty driven exploration methods capture epistemic uncertainty in state visitations [1], transition dynamics [2], Q-functions [3] or value functions [4]. However, we highlight our contribution as the first attempt to extract uncertainty from human feedback through learned reward functions.
>
> We emphasize that our motivation of RUNE, i.e. epistemic uncertainty over learned reward functions, is to develop human-guided exploration in preference-based RL. The ensemble of reward functions learned from human feedback is the only medium that communicates between RL agents and human teachers, and delivers information from human feedback. As we expect RL agents to achieve behaviors well-aligned with human intents, efficiently extracting useful information from human instructions is important to drive agent learning. RUNE encourages visitations of states and actions more uncertain with respect to human feedback, which provides a different perspective towards exploration in RL from uncertainty in state visitations or transition dynamics. Indeed, in our experiments, we found that capturing uncertainty of human feedback is more effective than other existing exploration methods. We hope this clarifies motivation and novelty in our approach and distinction from previous exploration methods.
>
> In addition, we provide an explanation about why naive exploration based on novelty in state visitations isn’t enough in preference-based RL. After states and actions are fully visited, there can still be high variance in predicted reward values of frequently visited states and actions because it receives fewer feedback queries. Such uncertainty in learned rewards may reflect instability in predicted reward values for RL policy training. We propose to explore these uncertainty states and actions with respect to learned rewards to showcase more these behaviors in environment interactions and in subsequent feedback sessions. This can improve reward learning as human preferences are tailored to reduce these uncertainties. We hope this explanation clarifies why exploration based on novelty in state visitations isn’t enough.
>
> ---
> [1] Burda, Yuri, et al. "Exploration by random network distillation." arXiv preprint arXiv:1810.12894 (2018).
>
> [2] Pathak, Deepak, Dhiraj Gandhi, and Abhinav Gupta. "Self-supervised exploration via disagreement." International conference on machine learning. PMLR, 2019.
>
> [3] Zhang, Yunzhi, Pieter Abbeel, and Lerrel Pinto. "Automatic curriculum learning through value disagreement." Advances in Neural Information Processing Systems 33 (2020).
>
> [4] Lee, Kimin, et al. "Sunrise: A simple unified framework for ensemble learning in deep reinforcement learning." International Conference on Machine Learning. PMLR, 2021.

---

> > ### Comment · Reviewer_x6bk · 2021-11-23
> > **Thanks for the response**
> >
> > Thank you for your detailed response, clarifications and revisions. This resolves many of the concerns I had, but there are still some limitations about the impact of the paper in my opinion. Nevertheless, I think that the paper definitely deserves a higher score than my original assessment.

---

> > > ### Author Response · Authors · 2021-11-24
> > > **Thank you for your additional comments**
> > >
> > > We are happy to hear that our rebuttal addressed your concerns well.
> > >
> > > Thank you again for the valuable suggestions and comments to add, which we will incorporate in the revision to strengthen our paper.
> > >
> > > If you have any remaining suggestions or comments, please let us know!
> > >
> > > Best, Authors.

---

> ### Author Response · Authors · 2021-11-22
> **A gentle reminder**
>
> Dear Reviewer x6bk,
>
> Thank you for your time and efforts in reviewing our paper.
>
> We kindly remind that the discussion period will end soon.
>
> We sincerely hope that our response and results of the supporting experiments successfully clarify your concerns.
>
> We just wonder whether we could have the last chance to address your further concerns or questions (if you have any).
>
> Thank you very much!
>
> Authors

---

### Decision · Program_Chairs · 2022-01-20

**Decision:**

Accept (Poster)

**Comment:**

This is a borderline paper. The scores were initially below the bar. The novelty of the work is limited and there are strong claims in the paper that should be revised. The authors can also do a better job in positioning their work with respect to the existing results. However, the authors managed to address several questions/concerns of the reviewers and convince them to raise their scores. I would strongly recommend the authors to address the rest of the reviewers' comments, especially those related to strong claims and connection to related work, and further improve their work in preparing its final draft.